# An Efficient Dehazing Algorithm Based on the Fusion of Transformer and Convolutional Neural Network

**DOI:** 10.3390/s23010043

**Published:** 2022-12-21

**Authors:** Jun Xu, Zi-Xuan Chen, Hao Luo, Zhe-Ming Lu

**Affiliations:** 1Wenzhou Mass Transit Railway Investment Group Co., Ltd., Wenzhou 325000, China; 2School of Aeronautics and Astronautics, Zhejiang University, Hangzhou 310027, China

**Keywords:** image dehazing, deep learning, convolutional neural network, Transformer, Transformer–Convolution fusion dehazing network (TCFDN)

## Abstract

The purpose of image dehazing is to remove the interference from weather factors in degraded images and enhance the clarity and color saturation of images to maximize the restoration of useful features. Single image dehazing is one of the most important tasks in the field of image restoration. In recent years, due to the progress of deep learning, single image dehazing has made great progress. With the success of Transformer in advanced computer vision tasks, some research studies also began to apply Transformer to image dehazing tasks and obtained surprising results. However, both the deconvolution-neural-network-based dehazing algorithm and Transformer based dehazing algorithm magnify their advantages and disadvantages separately. Therefore, this paper proposes a novel Transformer–Convolution fusion dehazing network (TCFDN), which uses Transformer’s global modeling ability and convolutional neural network’s local modeling ability to improve the dehazing ability. In the Transformer–Convolution fusion dehazing network, the classic self-encoder structure is used. This paper proposes a Transformer–Convolution hybrid layer, which uses an adaptive fusion strategy to make full use of the Swin-Transformer and convolutional neural network to extract and reconstruct image features. On the basis of previous research, this layer further improves the ability of the network to remove haze. A series of contrast experiments and ablation experiments not only proved that the Transformer–Convolution fusion dehazing network proposed in this paper exceeded the more advanced dehazing algorithm, but also provided solid and powerful evidence for the basic theory on which it depends.

## 1. Introduction

In many practical applications of computer vision, image restoration [1] plays an indispensable role. When acquiring images through various devices (such as cameras), a variety of factors, including blurring, noise and weather, among others, can affect image quality. In order to reduce or avoid the impact of these additional disturbances on subsequent advanced computer vision tasks, image restoration is often required as a preprocessing step of advanced computer vision tasks. Some advanced computer vision tasks, such as object detection, semantic segmentation, medical image analysis, etc., depend on the accuracy of image restoration. Image dehazing [2] is one of the important research fields in image restoration.

Haze is a traditional atmospheric phenomenon [3]. In hazy weather, the visual features of natural scenes will be seriously affected. This is because when hazy weather occurs, there are a large number of dust, smoke or other dry particles in the atmosphere, which constantly absorb and reflect light, leading to the degradation of visual quality.

The influence of haze on image quality results in a series of problems. In the field of ground photography, because light penetrating dense atmosphere is necessary for shooting distant objects, haze causes inaccurate estimates. In the same way, haze has a significant impact on some advanced computer vision tasks. Many advanced computer vision tasks require images with clear visibility as input; however, degraded images taken in hazy weather often cannot meet the requirements. This has the most serious impact on outdoor target recognition systems, outdoor automatic monitoring systems and intelligent driving vehicles. Therefore, in order to meet the requirements of high-level computer vision tasks, it is necessary to design an effective dehazing algorithm to restore the original colors and details to images with visibility degradation.

In recent years, with the development of deep learning [4] and the success of various network structures (convolutional neural networks, etc.) in advanced computer vision tasks, more and more researchers have begun to combine deep learning with image processing [5]. Because of their incomparable advantages over traditional methods, the powerful tools and diverse solutions of deep learning are very suitable for solving problems such as image dehazing.

Image dehazing methods can be divided into four categories according to their inputs: (1) multiple image dehazing, (2) dehazing based on polarization filter, (3) image dehazing using additional information (such as depth or geometric information), and (4) single image dehazing [6].

The dehazing method based on multiple images solves the dehazing issue by obtaining the changing atmospheric state from multiple images. In other words, it needs to wait until the atmospheric state or haze concentration changes. Therefore, this method is impractical for applications in the real world. Dehazing based on a polarization filter eliminates the need for real atmospheric state change. In this method, different filters are applied to different images to simulate changing weather conditions. However, only static scenes are considered when using the polarization-filtering-based haze removal method. Therefore, this method is still not suitable for real-time dynamic scenes. In order to solve the shortcomings of these methods, some researches have proposed dehazing techniques that use only the depth information or estimated scene 3D model information of a single image. However, it is usually very difficult to obtain additional information about the single image [7].

Our research mainly focuses on single image dehazing methods.There are two main methods to remove haze from a single image, i.e., the methods based on prior information [8,9,10,11,12] and the methods based on deep learning [13,14]. In this paper, a novel Transformer–Convolution fusion dehazing network is proposed, which further improves the dehazing ability of the network by integrating the global modeling ability of the Swin-Transformer and the local modeling ability of a convolutional neural network with adaptive fusion.

The contributions of this paper are as follows:

(1) A novel Transformer–Convolution hybrid layer is proposed, which aims to improve the dehazing ability of the network by using the global modeling characteristics of Transformer and the local modeling characteristics of convolution.

(2) The adaptive fusion mechanism is used to perform a learnable fusion of the output results of the Swin-Transformer and optional convolutional blocks.

(3) Compared with the simple residual blocks, we use two different convolutional blocks that are more suitable for image dehazing tasks to improve the ability of the network to extract and reconstruct features at different stages.

## 2. Related Works

### 2.1. Dehazing Methods Based on Prior Information

The method based on prior information mainly estimates the medium transmittance and atmospheric light intensity to describe the formation of haze by using some statistics of haze images. The final clear image is obtained by solving the atmospheric scattering model. This kind of method was widely used in the research of early image dehazing. Fattal [8,9] realized haze removal by analyzing reflectivity under the assumption that the medium transmittance and surface shadow are not locally related [8]. Observing that clear fogless images have higher contrast than haze images, Tan proposed a dehazing method to maximize local contrast [9].

He et al. proposed a dark channel prior (DCP) [10]. This method is based on the observation of statistical data of haze-free images. It is found that in most local image blocks except the sky, at least one channel has very low intensity values at some pixels, even close to 0. Using this prior information, we can restore a haze image to a haze-free image.

Unlike the use of local prior information, Berman and Avidan proposed a non-local color prior (NCP) [11]. They found that the color of a haze-free image can be well approximated by hundreds of different colors. Each different color is clustered. These colors form a tight cluster in the RGB space and are represented as a line, called the haze line. These haze lines can be used to estimate scene depth and demist images.

Zhu et al. put forward a color attenuation prior (CAP) [12] to recover the depth information by creating a linear model on the local prior to realize the estimation of the medium transmittance.

However, it is difficult to realize parallel acceleration due to the need to calculate various complex statistics from the image by manually selecting prior information, which leads to time-consuming dehazing of a single image. In addition, because the manually selected prior information does not fully conform to the mode of haze image generation, the restoration effect of this kind of method has certain limitations in both objective indicators and subjective evaluation. Due to the different choices of the prior information, incomplete dehazing or color distortion often occur.

### 2.2. Dehazing Methods Based on Deep Learning

Although the haze removal methods based on prior information have been successful to varying degrees, their performance is essentially limited by the assumptions adopted or the accuracy of the prior to the target scene [13], and the incomplete haze removal or color distortion often occur due to different choices of prior information. With the success of deep learning in some advanced computer vision tasks (image classification, image recognition, etc.), more and more researchers have begun to apply it to low-level computer vision tasks, such as image dehazing, and have made greater progress compared with the prior-based dehazing methods.

Cai et al. proposed a trainable end-to-end network DehazeNet [14] in 2016, which is used to estimate the medium transmittance and apply deep learning to image dehazing. They use neural networks to estimate the medium transmittance, which is more accurate than traditional methods.

In the same year, Ren et al. proposed an image dehazing method using a multi-scale convolutional neural network [15]. The algorithm is composed of a coarse scale network and a fine scale network. The coarse scale network can predict the overall medium transmittance map based on the entire image, while the fine scale network can refine the results locally. Compared with traditional methods, this method not only improves the speed of the image dehazing processing, but also greatly improves the quantitative analysis of the synthetic haze image dataset and the visual evaluation of real haze images.

With the success of the algorithm proposed by Cai, Ren and others, researchers began to shift their attention from the previous artificial selection to research on an image dehazing algorithm based on deep learning.

Li et al. proposed the integrated dehazing network AOD-Net [16]. AOD Net does not estimate the medium transmittance map and atmospheric light separately as the previous model did, but directly generates a haze-free image through a lightweight convolutional neural network. This novel design makes AOD-Net easy to embed in other models (such as Faster R-CNN), thus improving the effectiveness of advanced computer vision tasks on haze images.

He et al. proposed the densely connected pyramid dehazing network (DCPDN) [17], which can simultaneously learn the medium transmittance, atmospheric light and dehazing. End-to-end learning is achieved by embedding the atmospheric scattering model directly into the network. Inspired by DenseNet’s [18] ability to maximize the flow of feature information at different scales in the network, DCPDN proposed a new edge preserving dense connection encoder–decoder structure to estimate the medium transmission. For atmospheric light, DCPDN does not use manual selection but uses U-Net [19] to estimate atmospheric light. In terms of loss function, DCPDN uses the newly introduced edge preserving loss function to optimize. In addition, in order to further integrate the mutual structure information between the estimated medium transmissivity map and the dehazing results, DCPDN proposes a joint discriminator based on the GAN [20] framework of the generated countermeasure network, which improves the details by judging the authenticity of the corresponding dehazing image and the estimated transmission map.

The above research is based on the atmospheric scattering model and supervised learning to achieve image dehazing. Obviously, it is a natural idea to transform a foggy image into a fogless one based on the physical model, but the introduction of the physical model into image dehazing may also have some drawbacks. Therefore, some researches focus on image dehazing independent of the atmospheric scattering model.

Ren et al. proposed a gated fusion network GFN [21] to restore clear images directly from foggy inputs. GFN relies on encoder and decoder and adopts a new fusion-based strategy. GFN obtains three inputs of the network from the original foggy image by applying white balance, contrast enhancement and gamma correction. The encoder is used to capture the context information of the input image, and the decoder uses the context information captured by the encoder to estimate the contribution of each input to the final deblurring result and generate a pixel-level confidence map. Through the confidence map, the three input images are gated and fused to obtain the final dehazing image.

Liu et al. proposed a grid dehazing network, GridDehazeNet [13]. GridDehazeNet consists of three modules: pre-processing, backbone network and post-processing modules. Compared with the three pre-processing methods of GFN manually selecting foggy images, the pre-processing module of GridDehazeNet can also be trained, so the input of the backbone network has better diversity and more relevant features. The backbone network uses GridNet [22] as the framework to implement a new grid network multi-scale estimation method based on attention mechanism, which effectively alleviates the bottleneck problem often encountered by traditional multi-scale methods. The post-processing module is used to reduce artifacts in the final output.

Hong et al. proposed a knowledge distillation and demisting network KDDN [23] based on heterogeneous task simulation. In KDNN, teachers are an existing automatic encoder network for image reconstruction. The process-oriented learning mechanism is used to train the demisting network, assisted by the teacher network. The student network simulates the image reconstruction task in the teacher network. In addition, KDNN designed a spatially weighted color channel attention residual block for the student image dehazing network, which is adaptive to the color channel level attention perceived by the learning content and adds more attention to the reconstruction of dense haze areas.

With the development of unsupervised learning, some researchers also combine it with image dehazing. Engine et al. proposed cycle-dehaze [7], which does not require the training of pairs of foggy and fogless images but learns the style transfer from foggy images to fogless images based on CycleGAN [24]. Cycle dehaze adds a perception loss function on the basis of CycleGAN to improve the quality of texture information recovery and generate a visually better haze-free image. Alona Golts et al. proposed Deep DCP [25]. Unsupervised learning is achieved by minimizing the dark channel prior (DCP) energy function. Instead of using synthetic foggy images, they use real-world foggy images to adjust network parameters by directly minimizing DCP.

Since 2021, the field of image dehazing has developed rapidly, and more and more algorithms have emerged that can achieve impressive results on public datasets. In 2021, Wu et al. proposed AECR Net [26], which is the first time comparative learning was applied to image dehazing. By minimizing the contrast loss function, the features of the network output results are close to the features of the fogless image and far away from the features of the foggy image to further enhance the dehazing performance of the model. They achieved excellent performance with limited parameters and proved that the proposed contrast loss function can bring further improvement to many previous networks. In 2022, Song et al. proposed DehazeFormer [27] and successfully applied the Transformer, which has been successful in many visual fields, to the field of image dehazing. They modified many details of the Swin-Transformer [28] to make it more suitable for image dehazing tasks. DehazeFormer proposed a total of five models from small to large, and its largest model ranks first among all the currently published models in test results of the public dataset.

## 3. Proposed Transformer–Convolution Fusion Dehazing Network

With the rapid development of deep learning in recent years, the effect of single image dehazing algorithm is getting better and better. The results obtained by many algorithms on public datasets tend to be saturated with the increase of the size of convolutional neural networks.

The appearance of Transformer not only makes a breakthrough in advanced computer vision tasks, but also accelerates the performance improvement of image dehazing methods. As mentioned earlier, the DehazeFormer [27] scheme proposed by Song et al. integrates Transformer into the image dehazing scheme and makes targeted modifications. Compared with a series of previous algorithms based on convolutional neural networks, it has made more objective improvements. This also shows that Transformer can not only achieve good results in advanced computer vision tasks, but also surpass convolutional neural networks in low-level image restoration tasks.

In view of the superior performance of Transformer in dehazing tasks and its rapid development trend, using Transformer as a part of the network structure can improve the network feature extraction and recovery capabilities. However, due to the use of long-distance attention mechanism, Transformer has certain limitations in its ability to model local details. Although the Swin-Transformer reduces this limitation, it still obtains the global representation of some large areas in the image when the network is shallow. The convolutional neural network (CNN), which has the characteristic of local connection, can be used for local modeling better than Transformer.

For images, adjacent areas are highly correlated in terms of color and material at the low level and semantics at the high level. For the task of image dehazing, adjacent pixels tend to be approximately equal in terms of medium transmittance and haze concentration. Therefore, this paper proposes a new dehazing network based on the fusion of Transformer and convolution, which uses the local characteristics of CNN and Transformer’s powerful representation ability and global modeling ability to improve the performance of single image dehazing.

This section first introduces the overall network structure of our Transformer–Convolution fusion dehazing network (TCFDN) and then further introduces the Transformer–Convolution hybrid layer used in the network.

### 3.1. Network Structure

The network proposed in this paper adopts the self-encoder structure, one of the classic generation models in the field of image restoration. The network structure is shown in Figure 1.

The self-encoder consists of an encoder and a decoder. First, the fogged image with input size (B,3,H,W) is filled with three pixels of reflection (*B* represents the batch size of the image, three represents the number of channels of the input image, *H* represents the height of the image, *W* represents the width of the image), and then the size of the 32 convolution cores is 7×7. The convolution kernel with a step size of 1 expands the channel for inputting foggy images from 3 (RGB) to 32, and the image size becomes (B,32,H,W), while keeping the image size unchanged. After that, there are two consecutive convolution kernels with a size of 3×3. The convolution check image with step size of 2, filling pixel of 1, number of 64 and 128, respectively, is downsampled twice to reduce the image width and height to one-fourth of the original, and the number of channels is expanded to 128. At this time, the image size is (B,128,H/4,W/4). After obtaining the image with expanded channel number and reduced resolution, the feature extraction and reconstruction (Transform refers to Transformer) are performed using the proposed transform attention convolution hybrid layer and transform variable convolution hybrid layer. Then, we use two convolution kernels with a size of 3×3. For the transposed convolution with a step size of 2, a filling pixel of 1, and the number of convolution cores of 64 and 32, respectively, the reconstructed features are continuously upsampled twice, so that the image size becomes (B,32,H,W). This model uses the same weighted fusion technology as [26] to fuse downsampled and upsampled information. Finally, there are three convolution kernels with a size of 7×7. The convolution kernel with a step size of one restores the haze free image. At this time, the image size is (B,3,H,W), and three is the number of RGB channels.

### 3.2. Transformer–Convolutional Mixed Layer

In this paper, a novel Transformer–Convolution hybrid layer is proposed, which aims to improve the dehazing ability of the network by utilizing the global modeling characteristics of Transformer and the local modeling characteristics of convolution. The structure of the Transformer–Convolution hybrid layer proposed in this paper is shown in Figure 2.

In the whole network structure, the input of the Transformer–Convolution hybrid layer is a tensor with the shape of (B,128,H/4,W/4). First, we use 128 1×1-sized convolution operations to aggregate the input tensors on the channel dimension while keeping the number of input channels unchanged, and then the output of the 1×1-sized convolution is copied into a tensor whose shape is also (B,128,H/4,W/4). The Swin-Transformer and the optional convolution block are, respectively, used to extract or reconstruct features. The output shape and input shape of the Swin-Transformer and the optional convolution block are the same. In this way, two tensors with the shape of (B,128,H/4,W/4) after feature extraction or reconstruction are obtained. The two tensors are fused through the adaptive fusion module to obtain a tensor with the shape of (B,128,H/4,W/4). Finally, a tensor with the shape of (B,128,H/4,W/4) is obtained through the 1×1-sized convolution to aggregate its information on the channel dimension, and the tensors with the shape of (B,128,H/4,W/4) are output. Therefore, when the input tensor passes through the Transformer–Convolution hybrid layer, its shape will not be changed after feature extraction or reconstruction using Transformer and the convolutional neural network at the same time. This also makes it possible to adopt multiple Transformer–Convolution hybrid layers of the same structure in the network to improve the network’s dehazing ability.

The authors in [29] proposed a similar structure for image denoising tasks to integrate the feature extraction and reconstruction capabilities of Transformer and convolution. Its model structure is shown in Figure 3.

Although the model proposed in this paper has some similarities with the model proposed in [29], there are still large differences between them. In [29], after 1×1-sized convolution, it is divided into two parts whose channel number is half of the input channel number. However, the model proposed in this paper directly copies the output after 1×1-sized convolution, which can make full use of the information of all channels. The proposed model in [29] is the same among different model individuals, and the RConv Block in each Swin-Conv Block is the same. The classical residual block in the convolutional neural network is used. However, the model proposed in this paper introduces an optional convolution block; that is, different convolution blocks can be used between individuals of different Transformer–Convolution hybrid layers, which makes the network have corresponding processing means in different feature extraction and reconstruction stages. In addition, this paper uses two convolution blocks, rather than residual blocks, which are more suitable for image dehazing tasks.

The Swin-Conv Block proposed by [29] simply splices the Swin-Transformer and RConv Block after obtaining their outputs. In this paper, an adaptive fusion module is used to fuse the output of Transformer and the convolution layer effectively.

#### 3.2.1. Adaptive Fusion Module

In this paper, the adaptive fusion mechanism proposed in [30] is used to fuse the output results of the Swin-Transformer and optional convolution blocks. The structure of the adaptive fusion module is shown in Figure 4.

First, we add the output of the Swin-Transformer and the output of the optional convolution block to gather the information of the two outputs. We use global pooling, two connected full connection layers and a sigmoid activation function to obtain the corresponding weight of each channel and adaptively fuse the output of the Swin-Transformer and the output of the optional convolution block. The adaptive fusion calculation formula is shown below.
(1)$$output=input_{trans}+input_{conv}+w_{trans}\cdot input_{trans}+w_{conv}\cdot input_{conv}$$
(2)$$w_{trans}+w_{conv}=1$$

Through this adaptive fusion mechanism, the network can simultaneously use the information learned by Transformer and convolution. Moreover, the learnability of channel weights also enables the network to automatically select the information that is more conducive to the final haze-free image reconstruction learned by the Transformer and the convolution during the training process.

#### 3.2.2. Optional Convolution Block

Compared with the simple residual block used in [29], this paper uses two different convolution blocks that are more suitable for image dehazing tasks to improve the ability of network feature extraction and reconstruction at different stages.

In this paper, FA Block and deformable convolution proposed in FFA-Net [31] are used as convolution blocks in the Transformer–Convolution mixed layer. Studies, such as [26,31], have proved that the channel attention mechanism and pixel attention mechanism used in FA Block and the deformability of convolution in deformable convolution are very suitable for image dehazing tasks.

After downsampling the image, six trans-attention-convolution-blending-layers and two Transformer-deformable-convolution-blending-layers were used. The Transformer-attention-convolution-blend-layer uses FA Block in the optional convolution block, while the Transformer-deformable-convolution-blend-layer uses deformable convolution in the optional convolution block. This asymmetric network structure also makes the network feature processing methods for different stages more diversified than using only a single convolutional block.

## 4. Experiment and Analysis

After the Transformer–Convolution fusion dehazing network was proposed, detailed experiments were carried out to verify its effectiveness. This section compares the proposed demisting network with other algorithms and evaluates the performance of the proposed demisting network from different perspectives, such as the objective test results and subjective evaluation on the synthetic dataset and the subjective evaluation on the demisting effect of real haze images. In addition, a series of ablation experiments were conducted to prove the effectiveness of each module in the proposed dehazing network.

This section first introduces some settings of the experiment. Secondly, the synthetic dataset used is introduced. Then, the objective test results of the proposed dehazing network on the synthetic dataset, the subjective evaluation on the synthetic dataset, and the subjective evaluation on the dehazing effect of the real haze images are introduced in turn. Finally, this section introduces a series of ablation experiments.

### 4.1. Experimental Settings

This paper used Python to implement the proposed Transformer–Convolution fusion dehazing network. The Adam optimizer was used to train the model, and the exponential decay rate $\beta_{1}$ and $\beta_{2}$ were set to 0.9 and 0.999, respectively. The batch size during training was 16, and the initial learning rate was set to 0.0002. The cosine subtraction strategy was used to dynamically reduce the learning rate during training. In terms of loss function, L1 loss function and comparison regular term CR proposed by [26] were adopted. The CR parameter settings were the same as [26]. During training, the input image was randomly cut into 256×256 image blocks, and then randomly flipped and rotated. It should be noted how we selected the best model during training. Generally, there were two cases. One case was that the learning rate remained the same. We checked the loss after each iteration. When the loss converged and the results on the training set were stable, we stopped the training process. The other case was that the learning rate gradually decreased as the training progressed. At this time, the loss convergence may be caused by the decrease in the learning rate, so it did not necessarily mean that the network had been trained to the best performance. In our paper, we used the second case to select the proper number of iterations when the loss converged and the results on the training set were stable.

### 4.2. Dataset

This paper mainly used synthetic datasets to evaluate the performance of the proposed Transformer–Convolution fusion dehazing network. In image dehazing tasks, the most widely used dataset is RESIDE [32]. RESIDE includes five subsets, namely, indoor training set (ITS), outdoor training set (OTS), synthetic objective testing set (SOTS), real world task driven testing set (RTTS) and hybrid subjective testing set (HSTS). Among them, ITS, OTS and SOTS are synthetic datasets and are also the most widely used training sets and test sets. ITS includes 13,990 pairs of foggy image—no haze image pairs, OTS includes 313,950 pairs of foggy image—no haze image pairs, and SOTS includes indoor test sets and outdoor test sets with 500 pairs of foggy image—no haze image pairs each. This paper selected ITS as the training set and the indoor part of SOTS as the test set. This article uses NVIDIA RTX 3090 to train 600,000 iterations on ITS.

### 4.3. Objective Performance Comparison on Synthetic Datasets

In this paper, a traditional dehazing algorithm DCP [8] and eight deep-learning-based dehazing algorithms [13,14,16,23,26,31,33,34] were selected for comparison. Training on ITS, testing on SOTS indoor test set, using PSNR, SSIM and NIQE as evaluation indicators, the results are shown in Table 1. Here, structural similarity index measurement (SSIM) is a new index to measure the structural similarity of two images. The larger the value is, the better. The maximum value is one. It is often used in image processing, especially in image denoising and comprehensively surpasses SNR (signal to noise ratio) and PSNR (peak signal to noise ratio) in image similarity evaluation. The design idea of NIQE (natural image quality evaluator) is to build a series of features used to measure image quality and use these features to fit a multivariate Gaussian model. These features are extracted from some simple and highly regular natural landscapes; This model actually measures the difference in the multivariate distribution of an image to be tested, which is constructed by extracting these features from a series of normal natural images.

It can be seen from Table 1 that although the traditional method DCP has certain dehazing ability, the dehazing result is the worst. The dehazing algorithm based on deep learning makes use of the advantage that the neural network can learn from a large number of data and has significantly improved the dehazing effect compared with the traditional dehazing algorithm. DehazeNet and AOD-Net both use the atmospheric scattering model for dehazing, while other deep-learning-based dehazing algorithms are all generative models and do not use the atmospheric scattering model. It can be seen that the dehazing algorithm based on the generation model has a further improvement compared with the dehazing algorithm based on the atmospheric scattering model. AECR Net, with its proposed contrast loss function, has achieved better results than other deep-learning-based dehazing algorithms. The hybrid Transformer–Convolution network proposed in this paper combines the advantages of local features and global attention mechanism through convolution and Transformer. The PSNR tested on SOTS indoor reached 37.62, and the SSIM reached 0.9910, surpassing AECR Net and other deep-learning-based dehazing algorithms. Thus, Table 1, as important objective evaluation evidence, verifies the feasibility of the Transformer–Convolution hybrid network.

### 4.4. Subjective Performance Comparison on Composite Datasets

In this paper, two representative dehazing results in the indoor part of SOTS were selected to compare the subjective dehazing performance of different algorithms. The classical traditional dehazing method DCP and two more prominent algorithms in recent years, MSBDN and FFA Net, were selected for comparison.

The dehazing results of various algorithms are shown in Figure 5. Here, (a) is for the haze images; (b–d) are the dehazing results of DCP, MSBDN and FFA-Net, respectively; (e) is the dehazing results of the Transformer–Convolution fusion dehazing network proposed in this paper, and (f) is the clear fogless images corresponding to haze images.

By subjectively evaluating the dehazing results of different algorithms, it can be found that the overall dehazing results of the traditional method DCP are darker due to the use of dark channel prior, and the dehazing results still contain haze residues (Figure 5b). For the deep-learning-based algorithm MSBDN, FFA-Net, the haze in the foggy image has not been completely removed (in the red box area in the first line of Figure 5c,d). In addition, MSBDN produces irregular artifacts at the edges of some objects (chairs in the red box in the second line of Figure 5c), and FFA-Net is dark in some white areas (walls in the red box in the second line of Figure 5d); However, the Transformer–Convolution fusion dehazing network proposed in this paper, due to the integration of the feature extraction and reconstruction capabilities of the Transformer and the convolutional neural network, is closer to a clear haze-free image in the dehazing results, and there are no residual haze and uneven edges.

Figure 5 also proves that the network structure proposed in this paper has superior performance in image dehazing tasks from the perspective of conforming more to human cognition.

### 4.5. Subjective Performance Comparison on Real Haze Images

In order to further evaluate the ability of the algorithm proposed in this paper to remove haze from real haze images, a real haze image was selected for testing. DCP, MSBDN and FFA-Net were also used for comparison.

It can be seen from Figure 6 that the traditional DCP method (Figure 6b) has a good effect for real haze images, but some clouds are identified as haze in the sky for removal, resulting in color distortion in some sky areas. At the same time, there is still a layer of white haze on the edge of green plants in the lower right corner; For MSBDN and FFA Net, there is no significant change compared with the original image (Figure 6c,d), which shows that there are still some limitations in processing real haze images. The results of this paper can not only ensure the color of the sky area is unchanged, but also remove the haze in non-sky areas, and there is no residual haze near the edge of the object.

Figure 6 illustrates from another point of view that even if the network proposed in this paper is trained on the synthetic dataset, it can still effectively dehaze the real haze image. This also proves the good generalization ability of the network proposed in this paper.

### 4.6. Ablation Experiment

Although the above subsections have made a horizontal comparison between algorithms from an objective and subjective perspective and verified the feasibility of the proposed Transformer–Convolution hybrid network, we still carried out a series of ablation experiments to prove that the proposed Transformer–Convolution hybrid layer is more capable of feature extraction and recovery than the Swin-Transformer and the convolutional neural network.

The following network configurations were used for ablation experiments:

Configuration 1: Replaced the Transformer–Convolution hybrid layer with the Swin-Transformer. That is, after four times of downsampling of the input image, eight Swin-Transformer blocks were used for image feature extraction and restoration.

Configuration 2: Replaced the Transformer–Convolution hybrid layer with its corresponding optional convolution block. That is, after four times downsampling of the input image, six FA Blocks and two deformable convolutions were used to extract and restore the image features.

Configuration 3: Transformer–Convolution fusion dehazing network proposed in this paper.

The experimental settings and the selection of training and test sets were the same as those in Section 4.1 and Section 4.2. The results of training on ITS and testing on SOTS indoor are shown in Table 2.

It can be seen from Table 2 that only Configuration 1 of the Swin-Transformer was used after downsampling. Due to the focus on a large range of self-attention and the lack of local feature extraction ability, the final PSNR tested on SOTS indoor was only 34.68; After downsampling, only Configuration 2 of the convolutional neural network was used. Although it has good local feature extraction and reconstruction capabilities, it lacks some global feature extraction and reconstruction capabilities. Although PSNR was improved compared with Configuration 1, it was only 35.65 in the end. The network proposed in this paper (Configuration 3), using adaptive fusion strategy, combines the advantages of the Transformer and the convolutional neural network, and finally has a significant improvement compared with Configuration 1 and Configuration 2.

The ablation experiments carried out in this chapter provide powerful data support for the Transformer–Convolution fusion dehazing network proposed in this paper and further verify the effectiveness of the network.

## 5. Conclusions

This paper first introduces the research background and significance of image dehazing on the basis of a large number of research studies on image dehazing. After that, this paper proposes a dehazing network that combines the popular Transformer and a convolutional neural network (CNN) to make up for the shortcomings of the Transformer and the convolutional neural network individually. In order to verify the effectiveness of the Transformer–Convolution fusion dehazing network, we carried out a series of comparative experiments, using objective evaluation and subjective analysis to verify the feasibility of the algorithm from different perspectives, such as synthetic image dehazing and real image dehazing. The results show that the new dehazing network surpasses many existing dehazing algorithms in terms of both objective evaluation indicators and subjective visual perception. Our scheme is better than the state-of-the-art AECR-Net by 0.5dB on average in terms of PSNR, and 0.001 in terms of SSIM, which can also be proved by subjective evaluation. Finally, ablation experiments were carried out to further verify the theoretical basis of the proposed dehazing network.

## Figures and Tables

**Figure 1 sensors-23-00043-f001:**
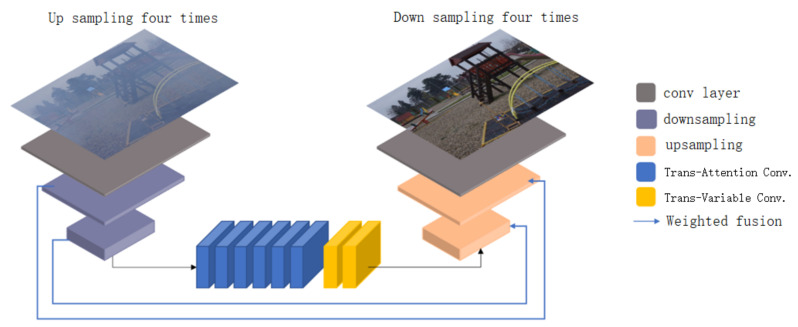
Our TCFDN network structure.

**Figure 2 sensors-23-00043-f002:**
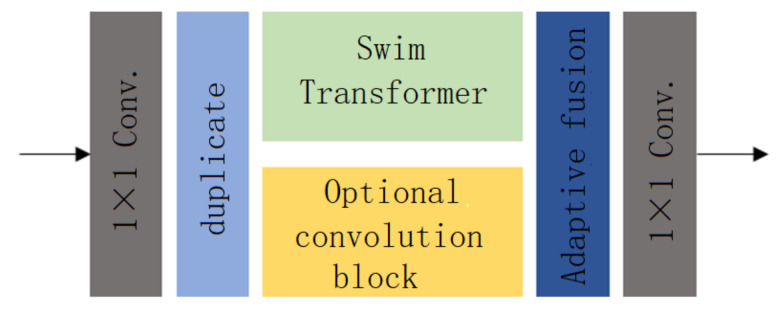
Transformer–Convolutional mixed layer.

**Figure 3 sensors-23-00043-f003:**
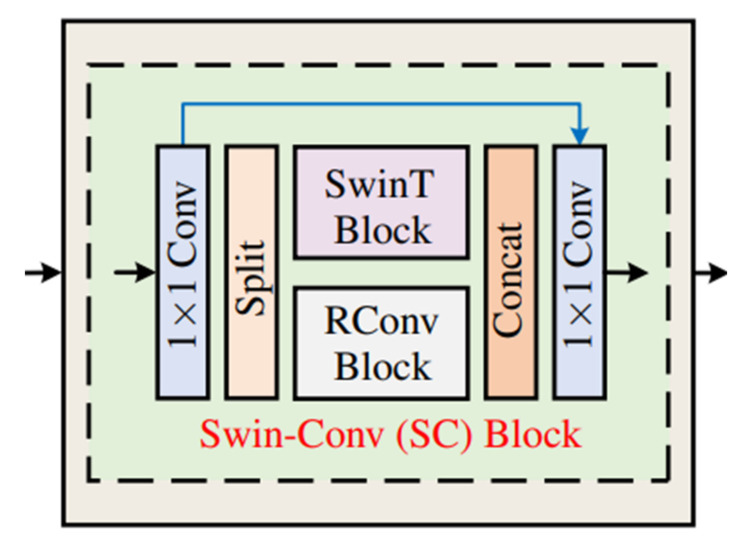
Swin-Conv block [29].

**Figure 4 sensors-23-00043-f004:**
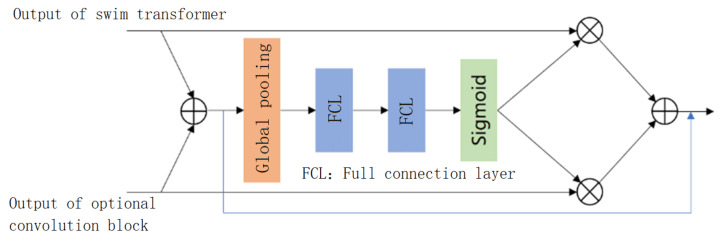
Adaptive fusion module.

**Figure 5 sensors-23-00043-f005:**
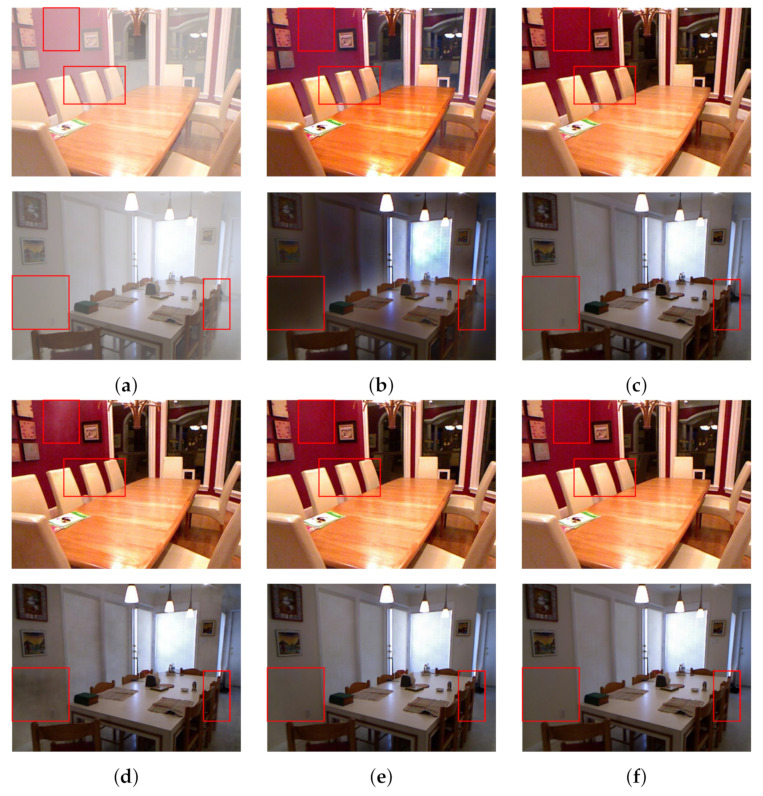
Comparison of dehazing results of synthetic haze images. (**a**) Haze image. (**b**) DCP [10] (PSNR: 15.09; SSIM: 0.7649). (**c**) MSBDN [33] (PSNR: 33.79; SSIM: 0.9840). (**d**) FFA Net [31] (PSNR: 36.39; SSIM: 0.9886). (**e**) TCFDN (PSNR: 37.62; SSIM: 0.9910). (**f**) Haze free image.

**Figure 6 sensors-23-00043-f006:**
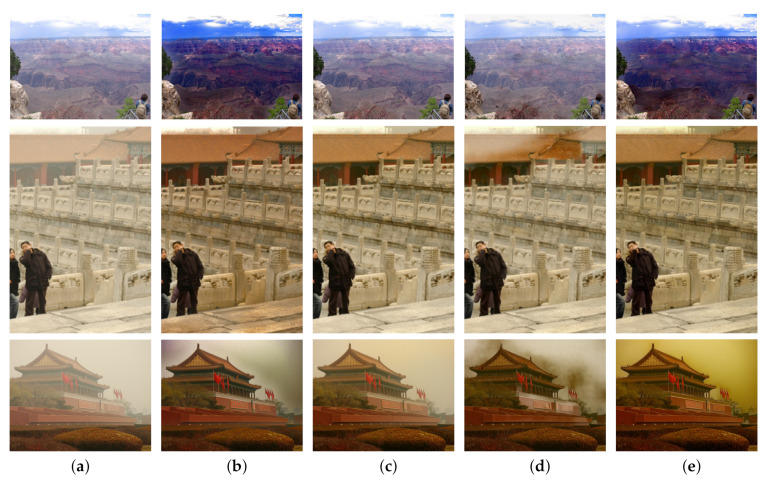
Comparison of dehazing results of real haze images. (**a**) Haze image. (**b**) DCP [10]. (**c**) MSBDN [33]. (**d**) FFA Net [31]. (**e**) TCFDN.

**Table 1 sensors-23-00043-t001:** SOTS objective performance comparison.

Algorithm	SOTS Indoor		
	PSNR↑	SSIM↑	NIQE
(TPAMI’10) DCP [10]	15.09	0.7649	3.1536
(TIP’16) DehazeNet [14]	20.64	0.7995	3.1709
(ICCV’17) AOD-Net [16]	19.82	0.8178	3.2243
(ICCV’19) GridDehazeNet [13]	32.16	0.9836	3.1800
(AAAI’20) FFA-Net [31]	36.39	0.9886	3.1796
(CVPR’20) MSBDN [33]	33.79	0.9840	3.1484
(CVPR’20) KDDN [23]	34.72	0.9845	-
(ECCV’20) FDU [34]	32.68	0.9760	-
(CVPR’21) AECR-Net [26]	37.17	0.9901	3.1730
Our TCFDN	**37.62**	**0.9910**	**3.2056**

**Table 2 sensors-23-00043-t002:** Ablation test.

Configuration	SOTS Indoor		
	PSNR↑	SSIM↑	NIQE
Configuration 1	34.68	0.9881	3.1811
Configuration 2	35.65	0.9882	3.2056
Configuration 3 (our network)	**37.62**	**0.9910**	**3.1925**

## Data Availability

The study did not report any data.

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
