# Peer review of "An Efficient Dehazing Algorithm Based on the Fusion of Transformer and Convolutional Neural Network"

_sensors, 2022, doi:10.3390/s23010043_

Round 1

Reviewer 1 Report

The authors proposed to use a combo of transformer and CNN layers to solve the task of image dehazing. Furthermore, A series of experiments are conducted to validate the advantage of the hybrid fusion between the two layers on publicly available dataset. The problem seems relevant, and the solution is technically sound. Nevertheless, several points need to be addressed:

- Authors can summarize the contribution of their work in the introduction section as bullet points.

- It is not specifically mentioned how did the author selected the best model during training. Early stopping strategy? monitoring validation loss? the last model of the training?

- The difference between the results presented in table 1 is very minimal. I would request the authors to check if the difference is statistically significant. 

- Since the performance advantage is very minimal as shown in table one, authors should refer to difficult image samples and show how the proposed method works better than compared methods.

- In addition to qualitative comparison, authors can add the PSNR and SSIM of the images displayed on figure 5.  

- although minor it's better to use words like image instead of picture in the manuscript

- It is advised that the authors should share the implementation codes to help other researchers working in a similar domain.

Reviewer 2 Report

1. Explain what is image dehazing first in abstract.

2. Introduction needs citations

3. remove auto hyphenation in MS word. Avoid using "algo-

rithms",

Com-
pared , etc

4. Fig 1 title is too generic, be specific

5. Figures can centred

6. Add a section for dataset details. number of images used

7. What is the importance of SSIM metric

8. Python 1.10.0. is old version. Just mention "Python"

9. Fig 6, "(e) Results of this paper". Mention model name

Reviewer 3 Report

Comments and Suggestions for Authors:

Lu et al. proposed a Transformer-convolutional fusion dehazing network structure to investigate the image recovery algorithm after single image dehazing. Based on the previous research, appropriate training and test sets are selected for experimental validation, and PSNR and SSIM metrics are selected for comparison with different algorithms to verify the effectiveness of the algorithm in this paper. Finally, the effectiveness of the algorithms in this manuscript is further verified by ablation experiments.

However, the test set images selected in the experimental session are not enough to verify the image dehazing recovery effect described in this paper, so it is strongly recommended to add two sets of less controversial experimental comparison images to verify the improved algorithm proposed in this paper.

1. Please add to the a priori information-based dehazing method the relevant research literature on the classical a priori validation of dehazing algorithms based on RGB values of haze images.

2. For the real haze image in the experimental comparison (Figure 6), the authors chose a controversial picture as a test effect map, can not well show the contrast between the DCP and the method proposed in the manuscript, for the distinction between clouds and fog is more subjective, can not be well compared to the fog effect. Two groups of real haze pictures should be added as the experimental group.

3. Please add more metrics about measuring image quality (e.g. NIQE, etc.) to Tables 1 and 2. Relying only on some of the previous criteria relying mainly on metrics such as PSNR, SSIM does not add convincingness in terms of superscore or other low-level visual dehazing tasks image evaluation, these metrics do not match our human eye senses.

4. Rewrite the conclusion part. The quantitative results are used to demonstrate the superiority of this manuscript algorithm over other algorithms.

5. Please make the implementation code about the Transformer-convolution fusion dehazing network publicly available on GitHub in order to check the feasibility of the experiment.

6. Inconsistent writing of proper nouns in the article, e.g. Trans-convolution, individual paragraphs (lines 277, 279) for trans-convolution, should be written consistently. Similar errors should be double-checked throughout the text. Also, please note the formatting consistency in line 433.

Round 2

Reviewer 2 Report

author made necessary changes based on review comments

Reviewer 3 Report

Agree to accept.